# Quantitative real-time analysis of the efflux by the MacAB-TolC tripartite efflux pump clarifies the role of ATP hydrolysis within mechanotransmission mechanism

Hager Souabni[1,2], William Batista dos Santos [1,2], Quentin Cece[1,2], Laurent J. Catoire[1,2], Dhenesh Puvanendran [1,2,4], Vassiliy N. Bavro [3] & Martin Picard [1,2 ✉]

Tripartite efflux pumps built around ATP-binding cassette (ABC) transporters are membrane protein machineries that perform vectorial export of a large variety of drugs and virulence factors from Gram negative bacteria, using ATP-hydrolysis as energy source. Determining the number of ATP molecules consumed per transport cycle is essential to understanding the efficiency of substrate transport. Using a reconstituted pump in a membrane mimic environment, we show that MacAB-TolC from *Escherichia coli* couples substrate transport to ATP-hydrolysis with high efficiency. Contrary to the predictions of the currently prevailing "molecular bellows" model of MacB-operation, which assigns the power stroke to the ATP-binding by the nucleotide binding domains of the transporter, by utilizing a novel assay, we report clear synchronization of the substrate transfer with ATP-hydrolysis, suggesting that at least some of the power stroke for the substrate efflux is provided by ATP-hydrolysis. Our findings narrow down the window for energy consumption step that results in substrate transition into the TolC-channel, expanding the current understanding of the efflux cycle of the MacB-based tripartite assemblies. Based on that we propose a modified model of the MacB cycle within the context of tripartite complex assembly.

[1] Laboratoire de Biologie Physico-Chimique des Protéines Membranaires, CNRS UMR 7099, Université de Paris, Paris, France. [2] Fondation Edmond de Rothschild pour le développement de la recherche Scientifique, Institut de Biologie Physico-Chimique, Paris, France. [3] School of Life Sciences, University of Essex, Colchester, UK. [4] Present address: Department of Cell Biology, Skirball Institute of Biomolecular Medicine, New York University School of Medicine, New York, NY, USA. ✉email: martin.picard@ibpc.fr

The rapid emergence of bacteria capable to resist multiple antibiotics has led to the development of multidrug resistance, that frequently puts in danger the lives of patients across the world. In extreme cases, bacterial strains have been shown to develop resistance against almost all known chemotherapeutic agents and antibiotics. In Gram negative bacteria, the impermeability of the double membrane is an important determinant of non-specific resistance[1]. The permeability barrier is formed by the combination of the action of protein assemblies, called efflux pumps, that actively export noxious compounds, and of a practically impermeable double membrane barrier composed of lipopolysaccharides, lipids, sugars and peptidoglycans that oppose to the passive diffusion of molecules into the cell[2].

Tripartite efflux pumps are composed of an inner membrane transporter, a periplasmic adaptor protein (PAP) attached to the inner membrane, and an extrusion channel inserted in the outer membrane. They assemble to allow for an efficient transport of drugs, dyes or detergents outside the cell, bypassing the periplasm. The first line of defense is composed of efflux pumps that belong to the Resistance-Nodulation-Cell-Division (RND) superfamily, the most-studied members of which include the AcrAB-TolC tripartite system from *E. coli* and the MexAB-OprM from *Pseudomonas aeruginosa*. Recently, the MacAB-TolC tripartite efflux pump from *Escherichia coli* raised additional and increasing interest[3]. In contrast to the RND efflux pumps that perform transport by consuming protons from the periplasm, MacB is a membrane protein from the ABC-superfamily that performs ATP-driven active translocation of substrates across the membrane. Substrate translocation occurs through ATP hydrolysis cycles concerted with conformational changes in the transmembrane domains (TMDs) where the substrate binds on one side of the membrane and is released to the other. In the assembled tripartite pump, a dimer of MacB recruits a hexamer of MacA (a PAP or membrane fusion protein, MFP), and a trimer of TolC (multifunctional outer membrane channel protein). MacB was the first antibiotic-specific ABC drug exporter experimentally identified in a Gram negative bacterium[4], initially shown to export macrolide compounds containing 14- and 15-membered lactones. Several structures describe this complex with atomic detail in resting and drug-transport states, suggesting that assembly and function of the pump are allosterically-coupled by a structural switch that synchronizes initial ligand binding and channel opening[5–9]. Biochemical studies showed that the ATPase activity of MacB is influenced by the presence of MacA and this was ascribed to its putative role in communicating the presence or absence of substrate in the periplasm to the nucleotide-binding and hydrolyzing domains (NBDs), located 100 Å away from the periplasm.

Overall, questions regarding the modes of assembly and requisites for optimal function have been well-characterized[10–13] but the puzzle of how exactly transport is coupled (i.e., how much ATP it takes to complete a full transport cycle), is still a matter of debate and controversies. This question is central to the mechanical description of the ABC-transporter cycle but is prone to pitfalls due to the hydrophobic nature of most ABC substrates that may spontaneously partition into membranes in a protein-independent fashion. As a consequence, kinetic measurements of both energy consumption and substrate transport are scarce in the literature[14–20] and such a measurement has not been achieved so far for a tripartite ABC-transporter.

We have decided to tackle the question of ATP:substrate coupling efficiency by the MacAB-TolC efflux pump from *Escherichia coli* upon its reconstitution into lipid membranes and to correlate ATP-hydrolysis with transmembrane transport.

## Results

**Overall principle of the assay.** Monitoring the activity of tripartite efflux pump from Gram negative bacteria is very challenging because they span the double membrane, and transport occurs from one environment to an other (see schematic representation Fig. 1a). Furthermore, the MacB family transporters have been demonstrated to specifically bind to[21], and to be modulated by phospholipids[22], which are also essential for the MacA-dependent activation of their ATPase activity[10,13], highlighting the importance of the adequate membrane environment for characterizing their function. In the past, we have designed procedures to mimic efflux by reconstituting the respective protein partners in lipid vesicles and monitor in vitro transport through a reconstituted pump[23–26]. Following similar lines, here we have designed a membrane-based system capable of mimicking the MacAB-TolC tripartite complex by using a lipid scaffold mimicking the two-membrane environment of the transporter and designing spectroscopic conditions allowing the monitoring of both ATP hydrolysis and substrate transport in real time. MacA and MacB were reconstituted into nanodiscs composed of POPC (1-palmitoyl-2-oleoyl-sn-glycero-3-phosphocholine) stabilized by a membrane scaffold protein (MSP, MSP1D1 or MSP1D1E3) that wraps around the hydrophobic core of the lipid discs[27] (see Methods section). Similarly, TolC was reconstituted into proteoliposomes composed of DOPC (1,2-dioleoyl-sn-glycero-3-phosphocholine) at a protein:lipid ratio such that, on average, one trimer of TolC was present per liposome. Tripartite complex formation was achieved by mixing MacAB-nanodiscs and TolC-proteoliposomes in a 1:1 molar ratio (Fig. 1b). Upon pump assembly, the MacAB nanodisc docks onto the tip of the periplasmic domain the liposome-incorporated TolC, forming an efflux-competent complex capable of transport of substrates from the buffering solution (equivalent to the periplasmic space) to the interior of the proteoliposome (representing the bacterial outer medium), at the expense of ATP-hydrolysis. As mandatory steps toward our goal to evaluate energy coupling, we have set up two independent procedures, described below, to monitor substrate transport (Figs. 1c and 2a, b) and ATP-hydrolysis (see Figs. 1d and 2c) in real time.

**QD-based monitoring of roxithromycin.** The main technical challenge of our study was to design a spectroscopic assay accounting for the real-time, quantitative, transport activity of the pump. To that end, we used quantum dots (QDs) as fluorescent sensors for the presence of antibiotics. Quantum dots (QDs) are nanometer-sized semiconductor crystals that exhibit excellent brightness and superior photostability compared to conventional fluorophores[28]. They are ideal for applications requiring high sensitivity, minimal interference with concurrently present probes, and long-term photostability. As additional asset, the fluorescence of CdTe QDs has been shown to be quenched in a dose-dependent manner in the presence of roxithromycin[29], a semi-synthetic macrolide antibiotic derived from erythromycin that is known to be a very good substrate of the pump[6]. As a first control, we verified that the range over which the concentration of analyte can be accurately monitored is compatible with our measurements. To that purpose, we measured QD fluorescence in the presence of increasing concentrations of roxithromycin. The ratio $F_0/F$ is plotted as a function of the quencher concentration (Stern–Volmer plot) to ascertain dynamic quenching is taking place (Fig. 2a, inset) and the variation is found to be linear, with a slope corresponding to a Stern–Volmer constant in perfect agreement with that measured by Peng et al.[29]. From all the above, it seemed appealing to take advantage of the specific fluorescence quenching of roxithromycin on QD probes to detect and quantify the passage of the antibiotic into the TolC acceptor proteoliposomes upon transport through the tripartite reconstituted pump (Fig. 1b). Indeed, when MacAB nanodiscs were

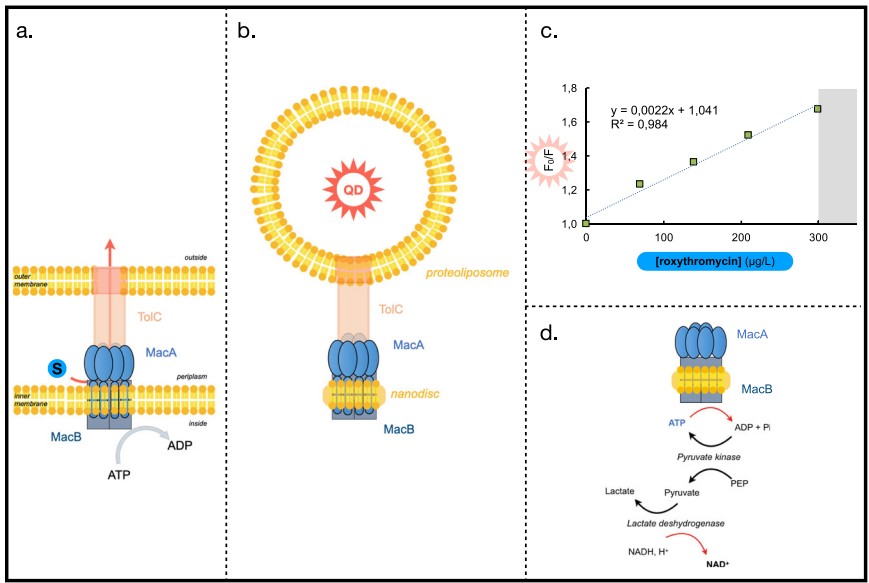

**Fig. 1 Rationale of the assay. a** Schematic representation of the efflux pump assembly in E. coli membranes. The MacAB-TolC pump encompasses the two-membrane barrier the Gram negative bacterium. A trimer of TolC, embedded in the outer membrane, protrudes into the periplasm and docks onto a hexamer of MacA that mediates the interaction with the periplasmic domain of MacB, which is a dimer. **b** Schematic representation of the in vitro assay. MacA and MacB, in lipid nanodiscs and TolC in proteoliposomes form the tripartite complex upon mixing and incubation. Transport through the whole efflux pump is studied by monitoring ATP hydrolysis and antibiotic transport from the solution, representing the periplasmic space to the interior of the liposome representing the exterior of the bacteria. **c** Stern–Volmer plot. Quenching of the quantum dots fluorescence in the presence of increasing concentrations of roxithromycin. Results are best described by the Stern–Volmer type equation: $F_0/F = 1 + K_{SV}[S]$, where F and $F_0$ are the fluorescent intensities of the QDs at a given roxithromycin concentration and in a roxithromycin-free solution, respectively. **d** NADH-coupled ATPase assay. Hydrolysis of ATP is coupled to the oxidation of NADH as a consequence of two consecutive enzymatic reactions. First, ADP generated upon ATP hydrolysis is regenerated to ATP by pyruvate kinase (PK) that catalyzes the transition of phosphoenolpyruvate (PEP) to pyruvate. Pyruvate is then reduced to lactate by lactate dehydrogenase (LDH), which concomitantly catalyzes the oxidation of NADH. Hence, decrease in ATP concentration is directly and stoichiometrically correlated to the decrease in NADH concentration.

preincubated with roxithromycin and TolC QD-entrapped proteoliposomes, a steep and biphasic decrease of QD fluorescence was measured as soon as ATP was added (Fig. 2a). At steady state, we measure a rate of increase of roxithromycin transport into the proteoliposomes equal to 1.33 mg transported per mL and per minute (see Fig. 2b), i.e., $1.6 \, \text{mM}^{-1} \, \text{min}^{-1}$. Deconvoluted to the quantity of protein present in the tube and to the estimated average internal volume of a liposomes of 100 nm diameter size, we estimate the rate of transport to about 65 nmol (of substrate) per mg (of MacB dimer) and per minute (measurement performed in triplicate, see Supplementary Fig. S3 for detail of the calculation). As mandatory controls, we also checked that no such quenching is observed when the experiment is performed in the presence of orthovanadate or in the absence of QD inside the TolC proteoliposome (see Supplementary Fig. S4b).

**ATP hydrolysis**. Kinetic studies of ATPase activity can be spectroscopically monitored using a Nicotinamide adenine dinucleotide (NADH)-coupled ATPase assay (see Fig. 1d) where decrease in ATP concentration is directly and stoichiometrically correlated to the decrease in NADH concentration[30]. Classically, the latter is deduced from variations in NADH specific absorbance at 340 nm. However, in our case, because MacB turnover number is known to be low (activity in the range of the tens of nmol(ATP).mg$^{-1}$.min$^{-1}$), this approach would require very high amounts of protein in order to reach ATP consumption levels sufficient for an observable decreasing of the absorbance signal. Attempting to measure ATP-hydrolysis under these conditions gave rise to optical artifacts (due to light scattering) which rendered interpretation impossible. Considering the fact that in diluted systems, fluorescence emission is proportional to the concentration of the fluorophore, we increased

the sensitivity of the assay by monitoring ATP-hydrolysis based on the variations of the intrinsic NADH fluorescence. As a preliminary step, we verified that indeed fluorescence is proportional to NADH over the concentration range used in our study (Fig. 2c, inset). Measuring NADH fluorescence as a function of time in the coupled-enzyme assay for MacAB nanodiscs (Fig. 2c) shows that ATP-hydrolysis is biphasic, as previously shown by Lin et al.[13]. The specificity of the ATPase stimulation was further confirmed by using an inactive MacB D169N mutant[10], containing substitution of a conserved residue forming part of the Walker B motif[31], involved in $Mg^{2+}$ coordination (Supplementary Material Fig. S4a).

Converting the slope of the fluorescence signal to the corresponding ATPase activity, it can be observed that following the initial "burst" associated with the ATP hydrolysis, the steady state specific activity of WT MacAB is equal to 21,9 nmol mg$^{-1}$ min$^{-1}$ (based on average of nine independent measurements, in the presence or in the absence of substrate). This value is somewhat lower than the previously reported[10,13], probably because it was stated in those study that care should be taken to very finely tune the molar ratio between lipids and detergent used during reconstitution reaction in order to reach highest level of MacAB ATPase activity[26].

## Discussion

While measurement of energy consumption is a rather straightforward and routine assay to determine the activity of ABC transporters, it does not always provide a reliable metric due to futile hydrolysis cycles possibly taking place, leading to apparent ATP-hydrolysis rates that are not always stoichiometrically coupled to substrate transfer. For example, eukaryotic homodimeric antigen-processing associated TAPL-complex undergoes ATP

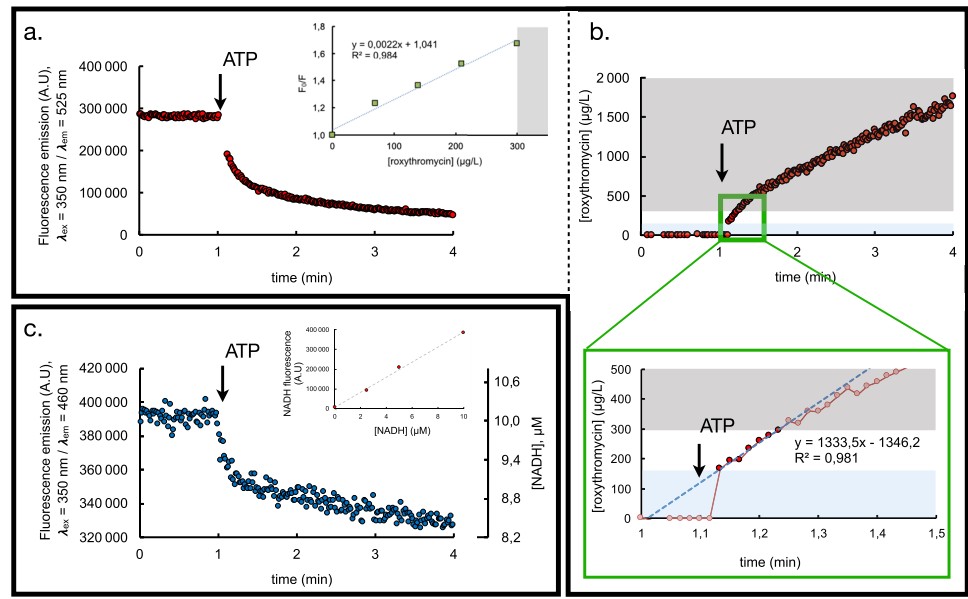

**Fig. 2 Real-time monitoring of roxithromycin transport and ATP hydrolysis within a reconstituted MacAB-TolC pump. a** CdTe quantum dot fluorescence is measured as a function of time after tenfold dilution of the MacAB / TolC complex into 20 mM Tris pH 8, 50 mM NaCl, 2 mM MgCl$_2$, addition of 7 µM roxithromycin (not shown here for the sake of clarity) and of 1 mM ATP (represented by the arrow). The MacAB nanodiscs (25 µg MacB, as standardized by SDS PAGE densitometry, see Supplemental Fig. S2) and TolC proteoliposomes were preincubated for at least 1 h at room temperature prior to their addition in the fluorescence cuvette, roxithromycin was added in the cuvette at least 10 min before addition of ATP. Changes due to dilution have been corrected for. inset: Quenching of the quantum dots fluorescence in the presence of increasing concentrations of roxithromycin shows that the fluorescence variation is proportional to the quantity of quencher up to a roxithromycin concentration of 300 µg/L. **b** Thanks to the above-described proportionality relationship, CdTe fluorescence quenching is converted into a quantity of roxithromycin transported as a function of time into the lumen of the liposome. The gray area represents the substrate concentration range for which proportionality does not hold. Values therein must not be taken into account. The green panel is a zoom of the region of interest. The blue area highlights the first fast phase of transport. **c** NADH fluorescence is measured as a function of time after tenfold dilution of the MacAB nanodiscs (c.a 10 µg MacB, as standardized by SDS PAGE densitometry, see Supplemental Fig. S1) into 20 mM Tris pH 8, 50 mM NaCl, 2 mM MgCl$_2$ containing the coupled-enzyme assay (see Methods for details) and addition of 1 mM ATP (represented by the arrow). Changes due to dilution have been corrected for. inset: linearity of NADH fluorescence over the range of concentration used in the assay. NADH fluorescence was measured with an excitation wavelength set at 350 nm and emission at 460 nm. Measurements were performed at room temperature (20 °C) in 20 mM Tris pH 8, 50 mM NaCl, 2 mM MgCl$_2$. The proportionality coefficient allowed the conversion of the fluorescence changes (primary y axis, left) into [NADH] variations (secondary y axis, right).

hydrolysis cycles that are highly uncoupled from substrate transport (20–100 ATP hydrolyzed per translocated peptide) consistent with a high degree of futile hydrolysis[15], while the heterodimeric TAP-complex shows strict coupling between peptide transport and ATP hydrolysis[14]. Whether futile, wasteful, expenditure of ATP is an intrinsic feature of ABC transporters or an artefactual consequence of the manipulation of detergent-solubilized proteins is a matter of debate[32].

Here, we show that at steady state, 1 mg of nanodisc-incorporated MacB transports ~65 nmol roxithromycin per minute, which in turn is associated with the hydrolysis of ~20 nmol ATP. This translates to at least three molecules of substrate being transported per single ATP-molecule hydrolyzed, a value that may reasonably be expected to be higher in vivo, as the interaction between MacAB and TolC under the tested conditions may not be perfectly stochiometric. Thus, under the conditions tested, MacAB transports substrate with very high coupling efficiency. Intriguingly, our data shows no connection between substrate presence and MacB ATPase activity. This suggests that the rate of ATP-hydrolysis may not be directly linked to substrate being present, and hence the conformational transitions of the periplasmic domains are not necessarily coupled to their association with the substrate. Indeed, similar conclusions have been derived from earlier functional studies of the binary MacA-MacB subcomplex, e.g., the effect of MacA on MacB ATPase stimulation in phospholipid bilayers, which showed that the presence of macrolides did not impact ATPase activity[10].

Further below we attempt to reconcile our findings with the currently available models of the assembly and function of the tripartite pumps formed with the participation of MacB-type ABC-transporters. Recent re-classification of the ABC-transporters taking into account their structural and topological organization has divided them into seven main family "types"[33]. Out of these, only two families are known to form tripartite assemblies, namely the group IV, that includes "classical exporters" (such as PrtD[34] and HlyB[35]) and the newly defined group VII which includes the MacB/FtsX type transporters[5,6,8], as well as the heterodimeric LolCDE lipoprotein sorting system[36], a representative of which is the well-studied MJ0796 transporter[37,38].

While the majority of the information on the ABC-transporter functional cycling, NBD-mediated ATP-binding, and ATP-hydrolysis related coupling is derived from type IV exporter group transporters that operate on alternate-access mechanism transporting their cargoes across the plasma membrane[35,39], the analysis of the MacB-like transporter structures and functional data[5,6,8] indicates that they operate on a radically different mechanical principle. One particular difference concerns the coupling of allosteric transitions between the TMDs and NBDs, which are mediated by the so-called coupling helices (CH) belonging to the TMD. While both type IV and type VII transporters have two CHs per protomer, in the former they are shared in a criss-cross fashion across the two NBD-protomers forming the functional transporter dimer, ensuring the cooperativity and direct coupling between the protomers, while in the latter (MacB-type) both CHs

are associated solely with their cognate protomers, suggesting difference in regulation and energy transmission. Indeed, in the MacB-family, the conformational changes in the NBD-associated with ATP-binding/hydrolysis which takes place in the cytoplasm, are allosterically communicated across the membrane to the substrate-binding periplasmic core domains (PCDs) in what has been dubbed "mechanotransmission"[5], although exactly which event provides the "power stroke" for the substrate efflux remains unclear.

The current paradigm of MacB-cycling is that the power stroke for transport is associated with the ATP-binding, while ATP-hydrolysis is used to reset the pump and provide directionality of transport[5]. This is in agreement with the results observed in type IV transporters e.g., recent structure of the lipid-linked oligosaccharide (LLO)-flippase PglK essential for asparagine-linked protein glycosylation in *Campylobacter jejuni*, in outward facing conformation, strongly suggests that the release of LLO to the outside occurs before ATP-hydrolysis and is followed by the closing of the periplasmic cavity of PglK[40]; while single-turnover assays for TmrAB in proteoliposome[41] indicate that a single ATP-binding stroke drives a stoichiometrically-coupled substrate translocation in a conformational switch that reorients the protein from an inner facing to an outward facing conformation. However, as pointed out by Moody & Thomas[42], any of the ATP-binding; NBD-dimerisation; ATP-hydrolysis; the release of the hydrolysis products; as well as any combination of these steps, could provide the power stroke during the ABC-transporter reaction cycle, and indeed, reconciling our results with this currently prevailing "fire bellows" model[5] suggests the existence of additional energy-coupling events contributing to the efflux cycle beyond the ATP-binding power stroke.

In the context of the tripartite assembly, this trans-membrane conformational coupling gives rise to the "fireplace bellows" model proposed by Vassilis Koronakis's group for the MacAB-TolC pump[5]. As mentioned above, in that model, it is postulated that ATP-hydrolysis is not used to directly transport the substrate across the membrane, but instead to mechanically reset the complex to its substrate-binding-competent apo-form following an efflux event, the power stroke for which is driven primarily by ATP-binding to the cytoplasmically-located NBDs. To understand fully the operation of tripartite assemblies, we need to take into account the additional levels of allosteric coupling and control associated with the presence of outer membrane components and adaptor proteins within them. Incidentally, to date, this effect could not be measured directly due to the technical difficulties associated with the reconstitution of functional assemblies into double membranes. One feature of the MacAB-TolC tripartite assembly is the "waiting room" cavity created at the interface between MacA-hexamer and MacB. This "waiting room" has been proposed to undergo a significant change in volume, associated with conformational changes in the periplasmic core domains (PCDs) of MacB upon its transition between nucleotide-free and ATP-bound forms, resulting in expulsion of the substrates that might be present there[5,6,8] (see schematic representation Fig. 3). Indeed, the existence of this "waiting room" cavity, along with the constant rate of ATPase activity, may account for the surprisingly large number of molecules transported per turnover reported here.

Given the central role given to the ATP-binding power stroke under the current "fire bellows" model, we were expecting to see a detectable disconnect between the transport, and ATP-hydrolysis. Indeed, previous experiments, on the MacB-related LolD/E system in the presence of vanadate[43], have demonstrated the possibility of a single cycle of extrusion by the transporter prior to ATP-hydrolysis. Interestingly, under the conditions tested in the current study, and contrary to the above results, MacAB-TolC pretreated with vanadate did not seem to be able to transfer

substrate into the TolC-liposome (Supplementary Fig. S4b). It has to be noted however, that the sensitivity of our studies does not allow us to conclusively rule out small quantities of substrate entering the TolC-liposome.

However, our results clearly demonstrate that ATP-hydrolysis and roxithromycin transport are biphasic and seemingly synchronous events, which is difficult to reconcile with above model (Fig. 3b).

One possible explanation for the observed behavior is that the initial substrate transport is facilitated by the positive substrate gradient across the TolC-containing membrane, which allows for a mixed-transport model, which apart from the ATPase driven transport, incorporates also an element of facilitated diffusion in the initial stages of the assay. However, the energy profile of the PAP-gate, which forms the exit of the "waiting room" chamber suggests that it functions as a one-way valve[8], which would to a large extent negate the concentration-gradient effects.

A second possible explanation for the observed behavior is indeed that the ABC-transporter cycle is in fact tightly controlled by the concentration of substrate and that the rapid decline in substrate concentration is leading to a corresponding decrease in ATPase activity to the basal level. This is an appealing idea, as it also provides a straightforward model of regulation of the pump in response to substrate presence, and also aligns itself with the earlier data on substrate-enhanced pump assembly[10,12,13].

One further intriguing possibility has been provided by the recent resolution of the structures of the Maintenance of PhosphoLipid Asymmetry (Mla) complexes[44], the structures of which show that MlaD-components form a hexameric "gasket-like" assembly on top of the dimeric MlaE-permease subunit of the MlaEF ABC-transporter in a similar fashion to the organization of the MacA-hexamer within the MacAB-subcomplex[45,46]. Furthermore, the TM-helices of different periplasmic-MlaD protomers were found to make distinct contacts with the TMD and Connecting Helix (CoH) of the MlaE permease, which is homologous to the CoH observed in the MacB-family. Mutation of these contact residues resulted in inactivation of the transporter function providing unexpected parallels to the MacAB-interaction, where TM-domains of MacA have been demonstrated to be essential for ATPase activity of MacB both in vitro and in vivo[10]. Such dependency was also observed in Gram-positive MacAB homologues, e.g., the ATPase activity of the Streptococcal Spr0694-0695 transporter reconstituted in proteoliposomes, was activated by full-length, but not by the N-terminally truncated version of the cognate PAP Spr0693[9]. Modelling of the MacAB-complex (Fig. 4), taking into account the PAP TM-helices, which were not resolved in the recent cryo-EM structures, indicates that similarly to Mla-complex they can make contact with the core of the transporter and this may provide an additional allosteric coupling of the ATP binding/hydrolysis to the periplasmic components of the complex, providing a mechanistic rationalization of the available functional data.

Taken together this suggests an alternative model of MacAB-TolC cycling, where ATP-hydrolysis provides additional energy to the transport cycle, complementing the central power stroke from ATP-binding (see Fig. 3b). This constitutes a critical difference between the current model and our modified scenario. While in the current model the substrate transit into the TolC-channel precedes the ATP-hydrolysis (which forms a separate, late stage of the cycle), in our model the substrate transit into the TolC-duct is concomitant with it.

As this ATP-hydrolysis is suggested to be under the control of the PAP protein[10,13] and as such is a particular feature of the tripartite assembly, it could not have been detected in the previous assays. Although circumstantial, the absence of detectable

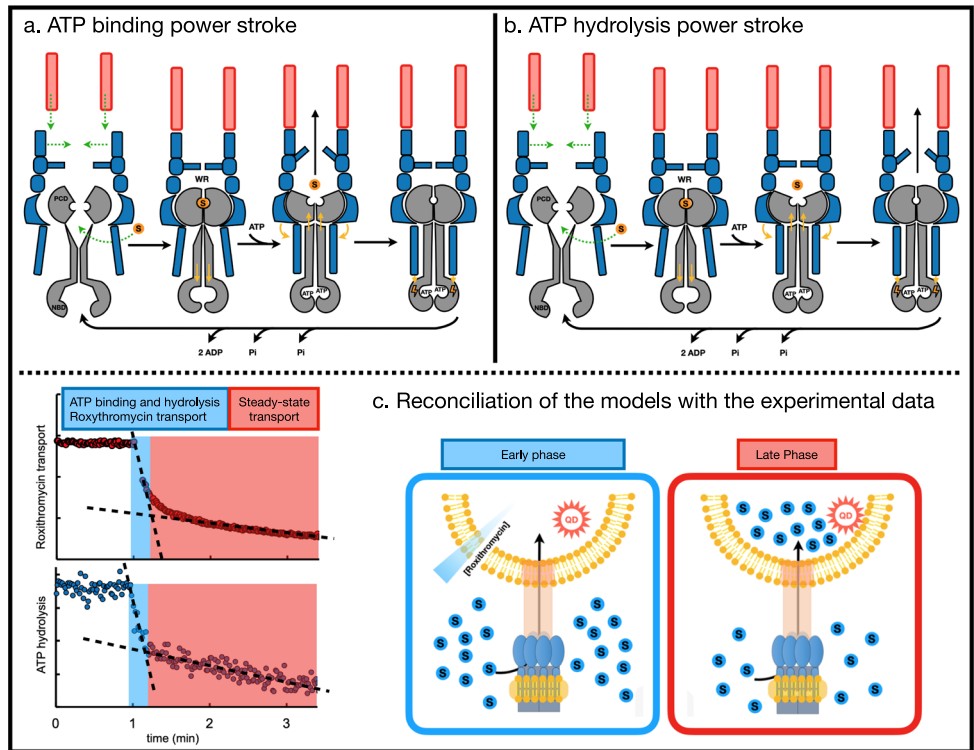

**Fig. 3 Two propositions for the "molecular bellows" mechanism of substrate-efflux by the MacAB-TolC tripartite efflux pump. a** The currently accepted model of the catalytic cycle proposed by Crow et al.[5], based on the fireplace bellows analogy, where ATP-binding induced changes within cytoplasmically-located NBDs are transmitted to the periplasmic domains of MacB (gray) via mechanotransmission-mechanism. These conformational transitions result is the substrates being directed into the TolC-duct (salmon), and they are prevented from flowing back into the periplasm by a "one-way valve" formed by MacA-loops (blue). The power stroke for transport is suggested to occur upon ATP-binding/NBD-dimerisation and that ATP hydrolysis allows to reset the pump to the substrate-binding capable inward-open state. Upon substrate engagement and full assembly of the tripartite complex (in a sequence of events that is still to be determined, hence the green dotted arrows), the reorientation of MacB's periplasmic core domains (PCDs) leads to rearrangement of its NBDs via downwards allosteric communication through its transmembrane "stalk"-helices. This reorientation of NBDs allows them to engage the ATP, leading to their dimerization, the power stroke of which is communicated back to the PCDs via the TM-helices, stabilizing the outward-facing open conformation of the PCDs. This in turn causes a reduction of the volume of the periplasmic "waiting room" (WR) cavity formed between the MacA-hexamer and MacB, resulting in a squeezing of the substrates present upwards toward TolC-channel. Crucially, in this scenario, the substrate transit into the TolC-channel precedes the ATP-hydrolysis, which appears to be a rate limiting step and is required for resetting of the tripartite pump to its initial substrate- and ATP-binding competent state. **b** A modified "bellows model" that rationalizes the assay data presented above. Which, while agreeing on the principal power stroke for mechanotransmission being derived primarily by ATP-binding/NBD-dimerisation, takes into account an additional contribution from the ATP-hydrolysis step, which provides vetting of the substrate transition to the TolC-channel. This step is likely associated with the PAP-dependent activation of the ATPase activity, which is subsequent to the large conformational changes in the PCD-domains of MacB associated with their transition to the outward-facing open conformation caused by the primary power stroke. The ATP hydrolysis in this scenario is concomitant with the transition of the substrate into the TolC-duct. The slow, rate-limited, resetting of the pump in this scenario would be associated with the release of the Pi and ADP. **c** The biphasic nature of the experimental results (left) could arise from a fast phase (colored in blue), which arises from the additional input provided by facilitated diffusion down the concentration gradient that exists at the start of the reaction, and from a slow phase (colored in red) that corresponds to steady-state transport upon its depletion (right).

substrate transfer in the presence of the vanadate, may indicate that this effect takes place relatively later during the cycle, which allows the opening of the "waiting room" gate concomitant with the PAP-mediated activation of the ATPase activity (Fig. 3b).

Thus, while our model agrees with the ATP-binding/NBD-dimerisation being the primary source of the power stroke that drives the efflux of substrates, our data indicates that an additional input of energy is provided by ATP-hydrolysis, via PAP-dependent mechanism, which accounts for the "late stage" allostery within the complex. Such allostery could be used for additional transmembrane communication between NBDs and the PCDs of MacB. Indeed, analogous energy storage/input from ATP-hydrolysis has been suggested to play a role in the Type IV transporter HlyB via gradual release of the inorganic phosphates post-hydrolysis, which are supposed to provide additional coupling to the TM-portion of the protein, which is required due to

the processive nature of the cycles needed to thread unfolded protein chain substrates[35,47].

In summary, the new procedure that we have described here to study MacAB-TolC might be applicable to other tripartite pumps that have been shown to pump roxithromycin. The applicability of our approach is not restricted to roxithromycin however, and can also be expanded to the study of pumps with different specificities, as recent reports demonstrate QDs fabricated for the detection of enrofloxacin, ceftiofur, doxycycline or chloramphenicol[48].

At present our methodology does not provide the time-resolution necessary to unambiguously discriminate between the above models of MacAB-TolC cycling and thus further functional studies, in particular targeting the kinetics of non-equilibrium pre-steady state (e.g., by stopped-flow fluorimetry), are needed to provide full kinetic parameters of the MacB cycle. However, by using a novel experimental set-up, this methodology allows to

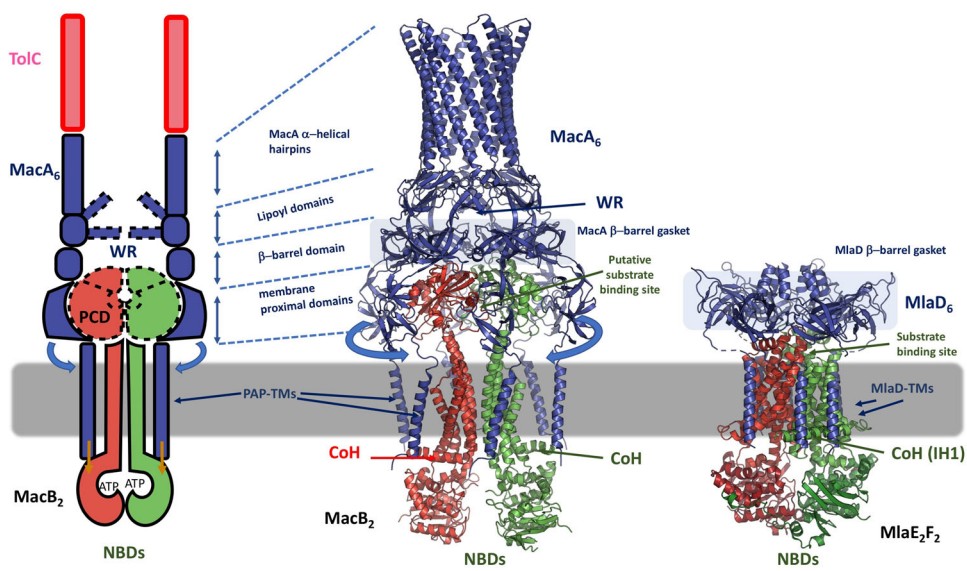

**Fig. 4 Communication across the membrane facilitated by the transmembrane helices of the periplasmic adaptor proteins and the TMD and connecting helices of the transporter may provide the additional long-range allosteric coupling that accounts for the additional energy input provided by ATP-hydrolysis power stroke.** Left panel, schematic representation of the MacAB-subcomplex, similar to Fig. 3 above, with key structural elements being shown. The TM-helices of MacB play a central role in mechanotransmission[3,5,8], however, the holistic view of the MacAB-transporter unit also requires taking into account the allosteric signaling by the MacA-component (indicated by the blue and orange arrows). While a large body of functional data exists that demonstrates the involvement of the MacA in the allosteric signaling and in particular the involvement of its TMs in communication of the ATPase-stimulatory signal to the dimerised NBDs of MacB[10,12,13], so far there has been no structural rationale for it. Recent structure solution of MlaFEDB complexes[45,46,52], (here, PDB ID 6XBD pictured, on the right), provide unexpected analogy in structural and functional organization with MacAB-subcomplex. Within these complexes, the TM-helices of the adaptor-protein MlaD form distinct contacts with the TMD and Connecting Helices (CoH) of the ABC transporter MlaEF. These contacts have been shown to be critical for ATPase function and coupling of substrate-extrusion. While currently available crystal and cryo-EM structures[8,9,53] do not contain structurally-resolved MacA TM-domains, modelling of the full-length MacA based on the available overall structure of MacAB-TolC complex (PDB ID 5NIK; central panel, cartoon representation), demonstrates that the TM-helices plausibly contact the coupling helix (CoH) and/or the intramembrane helices of MacB, which could provide a mechanistic explanation to the requirement of these MacA TM-segments for activation of ATPase-activity of MacB[10]. The TolC and MlaB subunits of the MacAB-TolC and MlaEFDB complexes are not drawn for clarity, in the central and the right panels, respectively. WR abbreviation indicates the "waiting room" cavity formed between the MacA hexamer and MacB PCD domains.

confirm the primacy of the substrate- and ATP-binding within the functional transporter cycle, and narrows down the time window of ATP-hydrolysis step to the final stage of the cycle, allowing to revise some of the key events within it.

Importantly, the modified model we propose provides a non-contradictory explanation of the results reported here and allows combining them with previous observations on the effects of PAPs on the tripartite assembly, which have not been integrated fully in the previous models.

## Methods

**Materials and reagents**. 1-palmitoyl-2-oleoyl-*sn*-glycero-3-phosphocholine (POPC) was purchased from Avanti Polar Lipids (USA), C12E8 (Octaethylene glycol mono-dodecyl ether, ≥ 98% purity, ref. P9825), Triton X-100 (BioXtra, ref. 9284), ATP (Adenosine 5′-triphosphate disodium salt hydrate, ref. A2383), ADP (Adenosine 5′-diphosphate ≥95%, ref. 01905), L-Lactic Dehydrogenase from rabbit muscle (ref. L1254), NADH (β-Nicotinamide adenine dinucleotide, reduced disodium salt, ref. N0786), Pyruvate Kinase from rabbit muscle (ref. P7768), PEP (Phospho(enol)pyruvic acid tri(cyclohexylammonium) salt, ref. P7252), roxithromycin (ref. R4394) and QDs (CdTe core-type, ref. 777935) were purchased from Sigma, SM2 Bio-beads were obtained from Bio-Rad. Ni Sepharose High Performance (His Trap HP) and Superose 6 10/300 column were purchased from GE Healthcare.

**Protein preparation**. MacA, MacB and TolC were expressed and purified as previously described in ref. [26] with slight modifications. The genes were individually cloned into pET22a and expression was realized in C43 strains in 2YT media. Protein over-expression was induced by adding 0.7 µM IPTG when cultures reached OD$_{600}$ = 0.4 (overnight induction at 20 °C). For solubilization, Triton X-100 was used for MacA and C12E8 for MacB and TolC. Membranes were diluted

in 20 mM Tris pH 8, 300 mM NaCl, 1 mM PMSF at 2 mg/ml (as determined by BCA) and detergent was added at 4 °C under gentle stirring at a 1:2 w/w ratio for 1 h for MacB and TolC, or at a 1:10 w/w ratio for 2–3 h for MacA.

The suspension was cleared by ultracentrifugation (1 h at 100,000 *g*) and the supernatant, to which 10 mM imidazole was added, was loaded at a flowrate of 0.5 mL/min onto a 5 mL Ni-NTA superflow resin column equilibrated with 20 mM Tris pH 8, 250 mM NaCl, 10% glycerol (w/v), 0.2% C12E8 or 0.2% Tx-100 (w/v). During the wash step, detergent was progressively adjusted/exchanged to 0.03% C12E8 for all proteins. Elution was performed using increasing imidazole steps (20, 40, 100, 250 mM) and proteins were eventually exchanged against Tris pH 8 20 mM, NaCl 150 mM, C12E8 0.03% on desalting columns (PD-10).

**Reconstitution of MacAB in nanodiscs**. POPC lipids were dissolved in methanol/chloroform (v/v), dried onto a glass tube under steady flow of nitrogen, followed by exposure to vacuum for 3 h. The lipid film was resuspended at 10 mM in the reconstitution buffer (Tris pH 8 20 mM, NaCl 150 mM, C12E8 0.03% or Tx 0.2% and Glycerol 20% (w/v) and then sonicated for 5 min (30 s pulse, 30 s pause) at 40 Watts (Branson Digital Sonifier® with probe).

Proteins in 20 mM Tris pH 8, 250 mM NaCl, 10% glycerol, 0.03% C12E8 were inserted into nanodiscs according to a previously established protocol[26] with the following modifications. MacB in C12E8 was mixed with POPC and MSP (MSPD1) at a final 2.5:90:1 MSP:lipid:MacB molar ratio. MacB and MacA were preincubated for 1 h at a 1:6 molar ratio and then mixed with POPC and MSP (MSPD1E3) at a final 2.5:90:1 MSP:lipid:MacA-MacB molar ratio. After 1 h incubation, detergent was removed by the addition of SM2 Bio-beads (quantity equal to 30 x the mass of C12E8) into the mixture shaken for 1 h at 4 °C. Suspensions were then centrifuged 30 min at 50,000 *g* and the supernatants were used extemporaneously.

**Reconstitution of TolC in proteoliposomes**. The POPC film was suspended at 10 mM in the same reconstitution buffer as above (Tris pH 8 20 mM, NaCl 150 mM, C12E8 0.03% and glycerol 20%) now supplemented with 20 µM CdTe QDs.

The suspension was heated for 10 min at 37 °C and was submitted to five freeze-thaw cycles. The liposomes were extruded through 400 nm membranes and through 200 nm membranes. C12E8 was then added to reach a 1:1 detergent/lipid ratio (w/w), reaching a saturating ratio for which it was previously suggested that subsequent Bio-beads SM2 addition leads to unidirectional insertion of the proteins. Proteins were added to the solubilized liposome suspension at a lipids/OprM 1:20 ratio (w/w). Detergent removal was achieved using Bio-beads at a Bio-bead/detergent ratio of 30 (w/w) at 4 °C overnight. Non-trapped QDs were removed on PD-10 columns.

**Protein quantification**. After purification and reconstitution into nanodiscs, samples were loaded on 12% acrylamide SDS-PAGE gels without boiling. After electrophoresis, the gels were stained with Coomassie Blue and digitally scanned. Densitometry was performed using ImageJ software. The linear regions in the densitometry profile were determined by measuring the density of standards with known protein amounts as shown in Supplementary Figs. S1 and S2.

**Measurement of the ATPase activity**. ATPase activity was determined at 20 °C with a coupled-enzyme assay in a PTI International type C60/C-60 SE fluorometer, with the excitation and emission slit widths set at 5 nm. The excitation wavelength was set at 350 nm and emission at 460 nm. Measurements were performed in a medium comprised of 20 mM Tris pH 8, 50 mM NaCl, 2 mM MgCl2, supplemented with 5 mM MgATP, 0.1 mg/mL pyruvate kinase (75 U/ml), 1 mM phosphoenolpyruvate, 0.1 mg/mL lactate dehydrogenase (150 U/ml), and an initial concentration of about 10 μM NADH.

**Measurement of the quantum dot fluorescence quenching**. Equilibrium fluorescence experiments were performed with a PTI International type C60/C-60 SE fluorometer with constant stirring of the temperature-controlled cell (20 °C) and with the excitation and emission slit widths set at 5 nm. The excitation wavelength was set at 350 nm and emission at 525 nm. Measurements were performed in 20 mM Tris pH 8, 50 mM NaCl, 2 mM MgCl$_2$.

**Modelling and molecular visualization**. Full-length homology models of MacA were created using I-TASSER[49], with specific template assignment based on the MacA protomers from PDB ID 5NIK (chain IDs D; E)[8]. MacA TM-helices were normalized using genometric restraints in Coot[50] and the resulting full-length MacA protomers were docked onto MacB to create the MacAB-subcomplex using the Coot SSM function[51], guided by PDB ID 5NIK. PyMOL Molecular Graphics System, Version 2.0 Schrödinger, LLC., was used to visualize the results.

**Statistics and reproducibility**. Data were collected on independent experiments. At least three independent experiments were run for each condition (three for roxythromycin transport, nine for ATP measurements). We repeated each experimental condition a minimum of three times. Each replicate required a new batch of protein (i.e., a new purification, reconstitution in ND/liposome). All replicates were sucessful. We have represented on Fig. 2 the most representative experiments.

**Reporting summary**. Further information on research design is available in the Nature Research Reporting Summary linked to this article.

## Data availability
The authors declare that the data supporting the findings of this study are available within Supplementary Data 1–4. Any remaining information can be obtained from the corresponding author upon reasonable request.

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

## Acknowledgements

We are grateful to Jean-François Gaucher, Dmitry Shvarev, Cédric Orelle, Jean-Michel Jault and Gregory Boël for their careful reading and useful comments on the paper. We also thank Jeanne Volatron from Everzom, for the nanoparticle track analysis. This work was supported by the French national research agency ANR (ANR-16-CE11-0001-01) and by the FRM (FRM-DBF-20160635738). H.S. was financed by FRM-DBF-20160635738 and Q.C. was financed by ANR-16-CE11-0001-01. We acknowledge the 'Initiative d'Excellence' program from the French State (Grant 'DYNAMO', ANR-11-LABEX-0011-01). D.P. and W.B. are recipient of MENRT fellowships from the French Ministry of Research and Education. V.N.B. wishes to acknowledge funding from BBSRC (BB/N002776/1).

## Author contributions

H.S. and M.P. designed the experiments. Q.C., H.S. and D.P. produced and purified the proteins. H.S, L.J.C. and Q.C. produced proteins in nanodisc. H.S. and W.B. carried out fluorescence measurements. V.N.B. provided structure-based analysis and modelling. V.N.B. and M.P. analyzed the data, wrote the discussion and produced the allosteric model of MacAB-TolC cycle. M.P. wrote the paper with inputs from all authors.

## Competing interests

The authors declare no competing interests.
