## [Peer Review File · Communications Biology]

Reviewers' comments:

Reviewer #1 (Remarks to the Author):

SUMMARY

The manuscript submitted to Communications Biology, titled 'Quantum dot probes for the quantitative study of drug transport by the MacAB TolC efflux pump in lipid scaffolds', by Souabni et al., introduces a new analytical method for the study of ABC tripartite efflux pumps—important bacterial membrane proteins involved in the phenomenon of antimicrobial resistance. The authors use a lipid scaffold that mimics the protein's native environment of the outer and the inner membranes and design a spectroscopic assay, employing quantum dots (QD), to monitor ATP hydrolysis and substrate transport in real-time. The presented method is applied to MacAB-TolC tripartite pump complex, suggesting high coupling efficiency between transport and energy consumption in the system.

GENERAL COMMENTS

The manuscript is well and clearly written, with adequate illustration and explanation of experimental methods used to ensure that the aim of disseminating a new methodology is achieved. If this is the aim of the paper, then the authors succeed in achieving it, however the manuscript, as a whole, presents a number of issues with respect to the its focus, general applicability as well as some of the experimental conclusions.

In their manuscript, the authors present a new method of studying the transport of antibiotics by bacterial efflux pumps, employing the phenomenon of fluorescence quenching in QD by the erythromycin derivative roxithromycin. As the method is based on the phenomenon observed in the case of this specific antibiotic, it is difficult to see how their analysis would be applicable to other transport substrates. That is unless the authors can cite other chemicals that exhibit this property and show that it can be exploited experimentally in a similar manner. Similarly, a suggestion whether this method might be applicable to other tripartite pumps that have been shown to pump roxithromycin (if any) might be welcome. I believe that the above issue also contributes to the certain 'preliminary' character of the study presented in the manuscript.

The authors titular aim is to introduce a new analytic method, but the introduction suggests that they specifically chose to 'tackle the question of ATP:substrate coupling efficiency by the MacAB TolC efflux pump'. This is then addressed experimentally by the authors' suggested methodology, which occupies the majority of the text. The study's focus is therefore somewhat diluted, and I would suggest that the authors consider restructuring the paper so that their message is stated clearly and conveyed more straightforwardly.

There are some issues with the experimental setup that I believe need to be addressed by the authors as well. A major problem area centres around the amount of available MacB, which forms the basis of the authors' calculation of coupling efficiency and therefore the major finding reported by the paper. Firstly, the total amount of protein is quantified only by a densitometric analysis of SDS-PAGE results. This in itself is not a problem, but this method should be supported by other protein concentration assays, to corroborate this very important value in their analysis. A similar issue pertains to the question of the total available quantity of MacB versus the quantity actually engaged in transport. The interaction of the MacAB component with the TolC might not be fully stoichiometric and therefore a complete coupling of MacAB to TolC cannot be assumed. The authors base their calculation on the total amount of protein added, which might not equal to the total

amount of protein involved in transport. Unless this issue is addressed and accounted for the calculation, the result can be seriously skewed. The interaction between TolC and MacAB has been shown to be in the nanomolar (Tikhonova et al., 2011; doi:2011;18:454–463), but is highly sensitive to experimental conditions, so should ideally be checked in a given setup.

Other two issues with the experimental setup are less serious but would need addressing as well. Firstly, the authors base their calculation on the average volume of the proteo-liposomes themselves taken from literature, therefore it would seem prudent to use a similar method to assess that value in their own experimental setup. Secondly, the authors say that their QD are in the nanometre range, but a more precise figure would perhaps help to alleviate the worry that their probes could be leaking from inside the proteo-liposomes through the TolC channel. This perhaps would happen if the channel was opened by the MacAB upon ATP hydrolysis and then that component was to dissociate off leaving the TolC open. Given what we know, that is unlikely, but cannot be excluded. The available diameter of the TolC opening has been reported as just slightly under 2 nm, as I am sure the authors know.

In their conclusion, the authors suggest that their results constitute a development on the “fireplace bellows” model suggested in a paper by Crow et al. (2018), cited in the manuscript. In the original study, it is suggested that the ATP hydrolysis and binding have differing roles: binding provides the power stroke and the hydrolysis ensures the directionality of transport. The authors of the manuscript suggest that the hydrolysis itself is the power stroke. They based this on their observation of the synchronous nature of the ATP hydrolysis and the substrate translocation, as evidenced by the graphs provided. Whilst this is an interesting and important observation, the authors need to acknowledge the limitations of their method, which does not have the resolution to record any sub-second phenomena that could be at play here.

SPECIFIC COMMENTS

Introduction

p.3.: Authors state that “In Gram negative bacteria, resistance is mostly due to the combination of protein assemblies, called efflux pumps, that actively export noxious compounds, and of an impermeable double membrane barrier composed of lipids, sugars and peptidoglycans that oppose to the passive diffusion of molecules inside the cell”. Please be advised that this overstates the importance of the two abovementioned mechanisms, specifically with respect to resistance borne by resistance plasmids, mutations in lytic enzymes, etc. Additionally, the authors’ given reference does not support the above phrasing. Please rewrite accordingly.

p.4.: In “subjected to pitfalls”, please consider writing “prone to pitfalls” instead.

Discussion

p.8.: Statement “MacAB is a very efficient transporter” does not offer any justification, or comparison, please elaborate.

Figures

Fig. 1a.: In the interest of clarity, please label cellular compartments and the membranes.

Reviewer #2 (Remarks to the Author):

The manuscript by Souabni et al. reports the substrate transport and ATP hydrolysis activities of tripartite drug efflux pump MacAB-TolC from *Escherichia coli*. This is a challenging issue, because to assemble the tripartite complex, it is necessary to prepare the double membrane environment representing the inner and outer membranes of Gram negative bacteria. To overcome this problem, the authors utilize the nanodiscs and the proteoliposomes. The experiments are well designed and findings provide important biological insight. The reviewer has the following comments.

Major points:

The substrate transport activity was measured with MacAB-TolC complex, but the ATP hydrolysis was measured with only MacAB. There is a possibility that TolC binding affect the ATP hydrolysis of MacAB, so if the authors have any data, please show and discuss about this issue.

In Figure S2, there is a non-negligible difference between MacB concentrations calculated from 10 ul ND AB and 20 ul ND AB. There must be a reasonable explanation for this, because this value is critical to estimate the rate of roxithromycin transport and ATP hydrolysis.

Minor points:

In Figure 1a, please change the labels of TolC, MacA and MacB to be colored like in Figure 1b, and adjust the position of label of TolC for clearly.

Figure 1c and Figure 2a inset are duplicated.

In Figure 3a legend, color of TolC is NOT "orange".

In Figure S3, "0.052 ug · ml⁻¹" should be "0.052 ug · ul⁻¹".

Specific comments to the reviewers' comments.

We would like to thank the two reviewers for their thorough analysis of our paper and the very helpful comments. We hope that we have addressed all their comments and criticisms in full.

Please find below the point-by-point response to **Reviewer #1** (highlighted in yellow in the corrected manuscript) and to **Reviewer #2** (highlighted in blue in the corrected manuscript).

Inspired by the reviewers comments and thanks to thorough discussions with colleagues, we have realized that our manuscript would benefit from additional analysis and from in-depth structural comparisons with related ABC transporters. Among most insightful discussions, those with Dr Vassiliy Bavro made a decisive impetus and benefiting from his expertise, the discussion allowed us to define what we believe is a more-robust model of the tripartite pump cycle, which we hope will be convincing to the reviewers. He has also provided a pseudoatomic model of the MacAB-TolC assembly which takes into account the transmembrane elements of MacA, which allowed to a new perspective onto the data. His contribution and involvement in improving the manuscript justifies his participation as co-author in the updated manuscript. Similarly, we also wish to add Laurent Catoire as co-author as he contributed significantly to the study thanks to the reconstitution procedure of MacAB in nanodiscs.

While the additional changes (highlighted in grey) correct and improve the manuscript beyond the specific comments of the reviewers, the structure and the message of the initial submission are fully preserved, yet extended and significantly improved. We would therefore like to thank the reviewers for their patience in allowing us the time-extension needed to address these changes to our overall satisfaction, which we hope is now reflected in the overall quality of the manuscript.

Reviewer #1:

SUMMARY

The manuscript submitted to Communications Biology, titled 'Quantum dot probes for the quantitative study of drug transport by the MacAB TolC efflux pump in lipid scaffolds', by Souabni et al., introduces a new analytical method for the study of ABC tripartite efflux pumps—important bacterial membrane proteins involved in the phenomenon of antimicrobial resistance. The authors use a lipid scaffold that mimics the protein's native environment of the outer and the inner membranes and design a spectroscopic assay, employing quantum dots (QD), to monitor ATP hydrolysis and substrate transport in real-time. The presented method is applied to MacAB-TolC tripartite pump complex, suggesting high coupling efficiency between transport and energy consumption in the system.

GENERAL COMMENTS

The manuscript is well and clearly written, with adequate illustration and explanation of experimental methods used to ensure that the aim of disseminating a new methodology is achieved. If this is the aim of the paper, then the authors succeed in achieving it, however the manuscript, as a whole, presents a number of issues with respect to the its focus, general applicability as well as some of the experimental conclusions.

>>> We thank the reviewer for his positive and constructive comments and address his concerns in the following.

In their manuscript, the authors present a new method of studying the transport of antibiotics by bacterial efflux pumps, employing the phenomenon of fluorescence quenching in QD by the erythromycin derivative roxithromycin. As the method is based on the phenomenon observed in the case of this specific antibiotic, it is difficult to see how their analysis would be applicable to other transport substrates. That is unless the authors can cite other chemicals that exhibit this property and show that it can be exploited experimentally in a similar manner.

>>> Indeed, the QD fluorescence quenching properties of roxithromycin provide a definitive asset for our particular purpose. Both reviewers mentioned that the manuscript understated that our procedure could be generalized to other transporters or efflux pumps. We have modified our wording and now focus on the fundamental aspect of the cycling by MacA-TolC.

Those readers that might be interested in the broader use of QD we refer to the following publication where QD are specifically synthesized in order to be compatible with the use of

various antibiotics (enrofloxacin, ceftiofur, doxycycline and chloramphenicol: <https://pubs.rsc.org/en/content/articlelanding/2020/ra/c9ra09894a#!divAbstract>.

This reference has also been added in the text (now reference [48]).

Similarly, a suggestion whether this method might be applicable to other tripartite pumps that have been shown to pump roxithromycin (if any) might be welcome.

>>> We thank the reviewer for the suggestion, and as it is indeed the case, we have now added such a sentence to the Discussion.

I believe that the above issue also contributes to the certain 'preliminary' character of the study presented in the manuscript.

The authors titular aim is to introduce a new analytic method, but the introduction suggests that they specifically chose to 'tackle the question of ATP:substrate coupling efficiency by the MacAB TolC efflux pump'. This is then addressed experimentally by the authors' suggested methodology, which occupies the majority of the text. The study's focus is therefore somewhat diluted, and I would suggest that the authors consider restructuring the paper so that their message is stated clearly and conveyed more straightforwardly.

>>> We thank the reviewer for their constructive input and agree that, in its initial form, our text suffered from a confusing thread and that the very title of the initial submission was in itself somewhat misleading.

Following the reviewer's suggestion, we have thoroughly revised the flow of the manuscript and have streamlined the message and discussion, which is now reflected in the changed title: "Quantitative real-time analysis of the efflux by the MacAB-TolC tripartite efflux pump clarifies the role of ATP hydrolysis within mechanotransmission mechanism". While the experimental section remained the same, we revised the conclusions and the key findings and proposed a modified model of the pump action, which also resulted in modification of key sentences in the Abstract and in the Discussion to make sure that the overall scope of our manuscript is now focused on the mechanistic insights of MacAB-TolC rather than on the methodological novelty of our approach.

We believe that the modifications provided have indeed improved the focus and the structure of the narrative in line with the expectations of the reviewer.

There are some issues with the experimental setup that I believe need to be addressed by the authors as well. A major problem area centres around the amount of available MacB, which forms the basis of the authors' calculation of coupling efficiency and therefore the major finding reported by the paper. Firstly, the total amount of protein is quantified only by a densitometric analysis of SDS-PAGE results. This in itself is not a problem, but this method should be supported by other protein concentration assays, to corroborate this very important value in their analysis.

>>> We thank the reviewer for raising this valid point. Indeed, while it may not have been immediately clear in the previous version of the manuscript, but we do indeed cross-validate the values obtained from densitometry of the effectively liposome/nanodisc-reconstituted proteins by an established colorimetric-titration method (BCA kit; Pierce), in addition to the measurement of the extinction coefficients during their purification, all of which give comparable values to those obtained from the densitometric analysis based on the SDS-PAGE mentioned. While not without caveat, we think that the procedure undertaken in our paper is sufficiently relevant, and precise, as it allows for the quantitation of the specific protein under investigation (i.e. MacA, MacB or TolC) within the context of the reconstituted tripartite complex, which is never 100% pure, due to the additional presence of the ND-scaffold protein.

A similar issue pertains to the question of the total available quantity of MacB versus the quantity actually engaged in transport. The interaction of the MacAB component with the TolC might not be fully stoichiometric and therefore a complete coupling of MacAB to TolC cannot be assumed. The authors base their calculation on the total amount of protein added, which might not equal to the total amount of protein involved in transport. Unless this issue is addressed and accounted for the calculation, the result can be seriously skewed. The interaction between TolC and MacAB has been shown to be in the nanomolar (Tikhonova et al., 2011; doi:2011;18:454–463), but is highly sensitive to experimental conditions, so should ideally be checked in a given setup.

>>> The issue raised by the reviewer is indeed critical and very challenging to tackle, however its relevance to the specific experimental set-up and the conclusions that we draw from it is limited. It is indeed very difficult to establish the exact figure for the engaged vs non-engaged complexes, as the reviewer rightfully suggests, however this is exactly why we focus on the substrate/ATP-consumption ratio instead.

While the assembly of the complex could be expected to be in the nanomolar range, it is impossible to ascertain whether 100% of the assembly taking place will result in a functional complex, and thus the specific activity of transport **could in fact be greater** than 65 nmol/mg/min. By contrast, the total amount of ATP hydrolysed per mg of MacAB₂ can be quantified precisely, giving a **lower limit** for the specific ATPase activity of 20 nmol/mg/min. In other words, the value of 3 molecules of substrate transported per ATP hydrolyzed will be true in the most favorable case where there is an exact stoichiometric interaction between all the MacAB and corresponding TolC present, and will be even larger if the interaction is not perfectly stoichiometric. We thank the reviewer for pointing this lack of precision and have now stated in the text that **“at least 3 molecules of substrate are transported per ATP hydrolyzed, a value that may be even higher if the interaction between MacAB and TolC were not perfectly stoichiometric”**.

Other two issues with the experimental setup are less serious but would need addressing as well. Firstly, the authors base their calculation on the average volume of the proteo-

liposomes themselves taken from literature, therefore it would seem prudent to use a similar method to assess that value in their own experimental setup.

>>> We thank the reviewer for his excellent comment! Indeed, as the reviewer suggests, providing an experimental value of the liposome luminal volume would be highly desirable. Following the reviewer's advice, we have tried hard to follow classical procedures such as that described by Oku et al. : A simple procedure for the determination of the trapped volume of liposomes. *Biochimica et Biophysica Acta (BBA) - Biomembranes* 691, 332–340 (1982) (doi: 10.1016/0005-2736(82)90422-9).

Unfortunately, these efforts were in vain due to their low sensitivity: considering the very small size of our liposomes, their residual trapped volume is so small that it is impossible to measure relevant and reproducible trapped volumes. To our great satisfaction, an alternative procedure allowed us to confirm the value proposed in our manuscript. Following the submission of the original manuscript, we have successfully implemented a new procedure for the quality control of our liposomes, using the Nanoparticle Tracking Analysis (NTA). This technique uses particle-by-particle light scattering to provide size information based on their Brownian motion (smaller nanoparticles move more quickly than larger particles, and scatter less light). In addition, in contrast to classical dynamic light scattering (DLS), each particle is sized independently, measured simultaneously and NTA's sizing principle is absolute. The readout of NTA-analysis consists of the absolute number of particles of a given diameter over a range of statistical intervals covering 5 nm – 1000 nm. By extrapolating the theoretical entrapped volume for each particle size interval and multiplying the latter by the corresponding number of (measured) particle we can calculate the value of the overall internal trapped volume. Most satisfactorily, we end up with a value almost identical to the one suggested in the initial submission (1,05 μ L with NTA / 1.064 μ L using the approximation from Mayer et al. *BBA* 1986) (doi: 10.1016/0005-2736(86)90302-0). We enclose the raw data that allowed us to obtain the NTA-based calculation of trapped volume for reviewer's consideration and would be grateful of their input on whether to include it as Supplementary material.

We have also provided a written explanation in the new text in the form of an expanded Legend for Figure S3.

Secondly, the authors say that their QD are in the nanometre range, but a more precise figure would perhaps help to alleviate the worry that their probes could be leaking from inside the proteo-liposomes through the TolC channel.

This perhaps would happen if the channel was opened by the MacAB upon ATP hydrolysis and then that component was to dissociate off leaving the TolC open. Given what we know, that is unlikely, but cannot be excluded. The available diameter of the TolC opening has been reported as just slightly under 2 nm, as I am sure the authors know.

>>> We use nanomaterials made from Cadmium Telluride core-type, COOH functionalized, Quantum Dots with an emission wavelength of 520 nm. They are described as

nanomaterials with a 4-6 nm diameter (<https://www.nanorh.com/product/cdte-core-type-quantum-dots-cooh-functionalized-fluorescence-%CE%BBem-520-nm-powder/>)

As can be deduced from the analysis of the 3D structure of TolC in its open form (a structure obtained by cryo-electron microscopy of a tripartite complex with AcrA and AcrB, two well-known efflux pump components comparable to MacAB, not shown here for the sake of clarity), it seems highly unlikely that a 4-6 nm nanoparticle can spontaneously leak out through the aperture of the TolC channel. A visual quantitative comparison of the TolC vs the expected nanoparticle sizes is provided below for illustrative purposes.

structure of open TolC(PDB: 5NG5, from Wang et al.)

Note that, in addition, we provide experimental evidence that there is no such leakage. Supplementary figure S4.b shows that when QD-entrapped TolC proteoliposomes are incubated with MacAB-ND pre-incubated (hence inactivated) by vanadate, the signal remains perfectly steady. If QD were prone to leak out of TolC, a continuous decrease would be observed (figure S4b, green circles).

In their conclusion, the authors suggest that their results constitute a development on the “fireplace bellows” model suggested in a paper by Crow et al. (2018), cited in the manuscript. In the original study, it is suggested that the ATP hydrolysis and binding have differing roles: binding provides the power stroke and the hydrolysis ensures the directionality of transport.

The authors of the manuscript suggest that the hydrolysis itself is the power stroke. They based this on their observation of the synchronous nature of the ATP hydrolysis and the substrate translocation, as evidenced by the graphs provided. Whilst this is an interesting and important observation, the authors need to acknowledge the limitations of their method, which does not have the resolution to record any sub-second phenomena that could be at play here.

>>> *We thank the reviewer for this accurate observation and we have sought to expand our structural interpretation of our data with the help of a structural biologist, inviting Dr Bavro to provide further comparative analysis with other ABC transporters. With his input we have reconsidered our conclusion and provide a modified model of the “fireplace bellows”, which is in agreement with the general conclusion of Crow et al., and also in agreement with the reviewer’s comment. Centrally important to our observation, and the key difference from the currently established model of MacAB-TolC operation is that the substrate transit into the TolC channel appears to be directly linked and synchronous to the ATP-hydrolysis. This does not rule out the key contribution of the ATP-binding/NBD-dimerisation as a power-stroke for the efflux, however it clearly demonstrates the presence of an additional, concurrent ATP-hydrolysis based input. In fact, we propose, that the ATP-binding/NBD-dimerisation is the primary contributor for the process, as witnessed by the cryo-EM and crystal-structures visualising the outer-facing conformations of the periplasmic domains of MacB in the absence of ATP-hydrolysis. In the newly modified version of our model, we have taken into account the contribution of the PAPs/MFPs to the functional cycle of the transporter and suggest that late-stage activation of the ATPase activity of MacB is allosterically coupled to the opening of the “waiting room” cavity formed by the hexameric PAPs/MFPs.*

While compatible with the previous model in its first part, our model suggests that there could not be a transfer of the substrate into the TolC-channel in the absence of MacB ATPase activity, something that is a direct consequence of the previous model, where the ATP-hydrolysis is a second, slow and possibly rate limiting step of the cycle.

This new model is described in further details in the updated Discussion section and presented in a heavily modified Figure 3, alongside with the currently prevalent model.

SPECIFIC COMMENTS

Introduction

p.3.: Authors state that “In Gram negative bacteria, resistance is mostly due to the combination of protein assemblies, called efflux pumps, that actively export noxious compounds, and of an impermeable double membrane barrier composed of lipids, sugars and peptidoglycans that oppose to the passive diffusion of molecules inside the cell”. Please be advised that this overstates the importance of the two abovementioned mechanisms, specifically with respect to resistance borne by resistance plasmids, mutations in lytic enzymes, etc. Additionally, the authors’ given reference does not support the above phrasing. Please rewrite accordingly.

>>> We have modified the sentence to state that this indeed refers to non-specific resistance and added that it is one of the primary mechanisms, but that the resistance is not exclusively due to the permeability barriers and efflux.

p.4.: In “subjected to pitfalls”, please consider writing “prone to pitfalls” instead.

>>> this sentence has been modified accordingly.

Discussion

p.8.: Statement “MacAB is a very efficient transporter” does not offer any justification, or comparison, please elaborate.

>>> The statement “very efficient transporter” has been replaced by “with very high coupling efficiency”.

Figures

Fig. 1a.: In the interest of clarity, please label cellular compartments and the membranes.

>>> Figure 1a has been modified accordingly. In addition to the label, we have also modified the representation of MacA, in order to account for its transmembrane segment.

Reviewer #2 (Remarks to the Author):

The manuscript by Souabni et al. reports the substrate transport and ATP hydrolysis activities of tripartite drug efflux pump MacAB-TolC from Escherichia coli. This is a challenging issue, because to assemble the tripartite complex, it is necessary to prepare the double membrane environment representing the inner and outer membranes of Gram negative bacteria. To overcome this problem, the authors utilize the nanodiscs and the proteoliposomes. The experiments are well designed and findings provide important biological insight. The reviewer has the following comments.

>>> We thank the reviewer for his constructive and positive comments.

Major points:

The substrate transport activity was measured with MacAB-TolC complex, but the ATP hydrolysis was measured with only MacAB. There is a possibility that TolC binding affect the ATP hydrolysis of MacAB, so if the authors have any data, please show and discuss about this issue.

>>> We thank the reviewer for pointing this out. Indeed, we have additionally performed ATPase activity experiments of MacAB in the presence of TolC. This experiment (now

presented in Supplemental figure 4) reveals that TolC induces a 20% acceleration of the ATP-hydrolysis (from 19 nmol /mg/min to 25 nmol/mg/min). This finding echoes a recent publication where we showed that in the related tripartite efflux pump MexAB-OprM from *Pseudomonas aeruginosa*, the activity of the inner membrane transporter (here energized by proton gradient, not by ATP) is also accelerated by the presence of the outer membrane protein OprM (Glavier, Puvanendran et al. Nature Communications 2020)(doi: <https://doi.org/10.1038/s41467-020-18770-5>). This interesting analogy between unrelated systems is beyond the scope of the present study and will be investigated further by our laboratory. We thank the reviewer for encouraging us to include these additional results in the modified version of the manuscript. We describe the corresponding experiment in a new Panel of Figure S4 (panel a, right).

In Figure S2, there is a non-negligible difference between MacB concentrations calculated from 10 ul ND AB and 20 ul ND AB. There must be a reasonable explanation for this, because this value is critical to estimate the rate of roxithromycin transport and ATP hydrolysis.

>>> The reviewer is correct to point out the discrepancy in the band intensity (hence MacB concentration), which is almost three times higher, while the quantity loaded on the gel is only twice as much. We confess that our only explanation for this discrepancy is the experimental stochasticity inherent to this kind of procedure (e.g. experimental error in the volume loaded, non-homogenous coloration by Coomassie Blue within the polyacrylamide gel). We acknowledge this weakness in the Figure Legend of Figure S3, but nevertheless firmly stick to our conclusion as if there was a systemic issue, it would be expected to proportionately affect both transport and ATP-hydrolysis as both of them depend on the protein quantitation performed with the same densitometric methodology, which should hence present the same bias. This hasn't been the case however.

Minor points:

In Figure 1a, please change the labels of TolC, MacA and MacB to be colored like in Figure 1b, and adjust the position of label of TolC for clearly.

In Figure 3a legend, color of TolC is NOT "orange".

In Figure S3, "0.052 ug · ml⁻¹" should be "0.052 ug · ul⁻¹".

>>> We thank the reviewer, all typos have now been corrected.

Figure 1c and Figure 2a inset are duplicated.

>>> The same inset is present in both cases on purpose: in Figure 1 to illustrate our overall methodology, in Figure 2 for the sake of interpretation and quantification of the raw data.

Additional modifications to the text.

Below we list the additional changes which we have incorporated into the manuscript beyond the cosmetic improvement.

At the end of the result section, we acknowledge that the measured ATPase activity is somewhat lower than the previously reported and provide a tentative explanation for this discrepancy.

Additional paragraph has been added to the discussion with structural description of the MacB and type IV ABC-transporters (e.g. HlyB) and we have expanded and clarified the mechanotransmission.

The role of ATP-binding and NBD-dimerisation is also critically reviewed and incorporated into the new functional “fire-bellows” model.

We have modified the text and legends of Figure 3 ; new model of MacAB-TolC cycling has been provided. We have removed the C, D panels of Figure 3 as we have modified the text, since we realized that the former left and right panel are applicable to both of the respective models of MacAB-TolC cycle proposed.

Significantly, we have now integrated the role of the PAP/MFP-components of the tripartite assembly into the model and have modified the role of the periplasmic “waiting room” suggested by Koronakis’ group (doi: 10.1073/pnas.1712153114) and suggested to play a role in creating a one-way valve (doi: 10.1038/nmicrobiol.2017.70).

Since the initial submission of the manuscript, structures of the MlaDEFB complex have been elucidated (Coudray *et al.*, doi: 10.7554/eLife.62518; Zhang *et al.*, doi: 10.1038/s41421-020-00230-5; Chi *et al.* doi: 10.1038/s41422-020-00404-6), which has provided us with an additional insight onto the possible allosteric coupling of the holo-transporter. With the help of Dr Bavro, we created a pseudo-atomic model of the full-length MacAB-subcomplex which has been included as a new Figure 4. This also covers structural comparison with the MlaEF-D subcomplex.

An additional section related to modelling and structure analysis has been added to the Materials/Methods. Further discussion has been provided to this effect in the main text.

These changes have been reflected in a modest modification of the Abstract, and a new title has been provided as stated above.

REVIEWERS' COMMENTS:

Reviewer #1 (Remarks to the Author):

The revised version of the Souabni et al. manuscript is a substantial reworking of their original submission to Communications Biology. The authors adjusted their title from 'Quantum dot probes for the quantitative study of drug transport by the MacAB TolC efflux pump in lipid scaffolds' to 'Quantitative real-time analysis of the efflux by the MacAB-TolC tripartite efflux pump 4 clarifies the role of ATP hydrolysis within mechanotransmission mechanism'. Supported by substantial changes to the main text, this represents a significant shift in the papers focus and the overall message it is conveying. In its original version, the manuscript was presented as a report on a novel technique for the study of antibiotic transport in the tripartite efflux system, using MacAB-Tol and antibiotic roxithromycin. By shifting the focus, the authors were able to concentrate on their most important finding, the elaboration of the existing "molecular bellows" model. This is an interesting and significant finding in the study of the tripartite efflux pumps.

The authors carefully considered mine and the other reviewer's remarks, producing a developed manuscript that includes additional experiments and shows the authors attention to received critique. Whilst the original submission felt perhaps slightly "episodic", after review, the manuscript delivers a coherent story. As all of my concerns have been properly addressed, I have no further general comments to offer. With regards to any specific comments, I would perhaps make a suggestion that the authors brake up their Introduction section into paragraphs, to improve readability.

Reviewer #2 (Remarks to the Author):

The authors have addressed my concerns. In addition, the extended discussion in the revised manuscript provide insightful suggestions about the molecular mechanism of substrate efflux by the MacAB-TolC tripartite complex. I don't have any major comments. I believe the manuscript can be published.

Minor Point:

In Figure 1b legend, "OprM" should be "TolC".

Specific comments to the reviewers' contribution.

We would like to thank the two reviewers for their time and in-depth examination of our paper. Their comments and suggestions have been very helpful and obviously contributed significantly increasing the impact of our conclusions.

We enclose a revised version that includes a final improvement (braking up of the Introduction section into paragraphs, as recommended by reviewer 1) and a final correction (“OprM” has been corrected for “TolC” in Figure 1b legend, as pointed out by reviewer 2).

We include below the latest reviewers' comments:

REVIEWERS' COMMENTS:

Reviewer #1 (Remarks to the Author):

The revised version of the Souabni et al. manuscript is a substantial reworking of their original submission to Communications Biology. The authors adjusted their title from 'Quantum dot probes for the quantitative study of drug transport by the MacAB TolC efflux pump in lipid scaffolds' to 'Quantitative real-time analysis of the efflux by the MacAB-TolC tripartite efflux pump 4 clarifies the role of ATP hydrolysis within mechanotransmission mechanism'. Supported by substantial changes to the main text, this represents a significant shift in the papers focus and the overall message it is conveying. In its original version, the manuscript was presented as a report on a novel technique for the study of antibiotic transport in the tripartite efflux system, using MacAB-Tol and antibiotic roxithromycin. By shifting the focus, the authors were able to concentrate on their most important finding, the elaboration of the existing "molecular bellows" model. This is an interesting and significant finding in the study of the tripartite efflux pumps.

The authors carefully considered mine and the other reviewer's remarks, producing a developed manuscript that includes additional experiments and shows the authors attention to received critique. Whilst the original submission felt perhaps slightly "episodic", after review, the manuscript delivers a coherent story. As all of my concerns have been properly addressed, I have no further genral comments to offer. With regards to any specific comments, **I would perhaps make a suggestion that the authors brake up their Introduction section into paragraphs, to improve readability.**

Reviewer #2 (Remarks to the Author):

The authors have addressed my concerns. In addition, the extended discussion in the revised manuscript provide insightful suggestions about the molecular mechanism of substrate efflux by the MacAB-TolC tripartite complex. I don't have any major comments. I believe the manuscript can be published.

Minor Point:

In Figure 1b legend, “OprM” should be “TolC”.